# URDFormer: Constructing interactive Realistic Scenes from Real Images via Simulation and Generative Modeling

## Abstract

Constructing accurate and targeted simulation scenes that are both visually and physically realistic is a significant practical interest in domains ranging from robotics to computer vision. However, this process is typically done largely by hand - a graphic designer and a simulation engineer work together with predefined assets to construct rich scenes with realistic dynamic and kinematic properties. While this may scale to small numbers of scenes, to achieve the generalization properties that are requisite of data-driven machine learning algorithms, we require a pipeline that is able to synthesize large numbers of realistic scenes, complete with "natural" kinematic and dynamic structure. To do so, we develop models for inferring structure and generating simulation scenes from natural images, allowing for scalable scene generation from web-scale datasets. To train these image-to-simulation models, we show how effective generative models can be used in generating training data, the network can be *inverted* to map from realistic images back to complete scene models. We show how this paradigm allows us to build large datasets of scenes with semantic and physical realism, enabling a variety of downstream applications in robotics and computer vision. More visualizations are available at: https://sites.google.com/view/urdformer/home

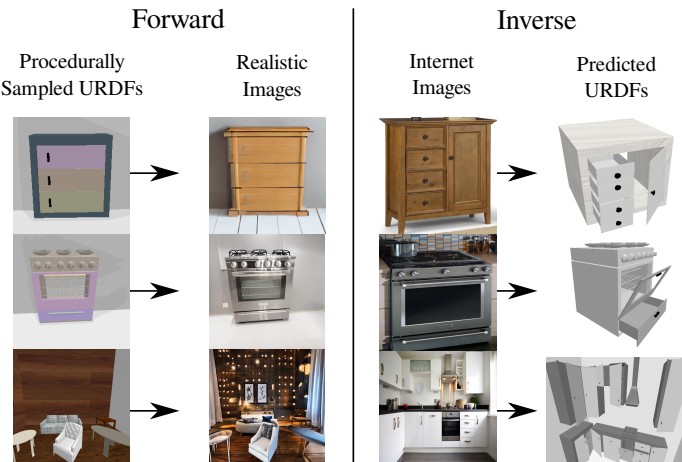

Figure 1: Our method uses generative models in a "Forward" process to produce structurally consistent realistic images from procedurally generated simulation content. We then use these generated simulation/image pairs to train an "Inverse" process that is able to estimate the underlying structure of diverse real-world images.

## 1 Introduction

Simulation has become a cornerstone of a plethora of applied machine learning problems - from the natural sciences such as physics, chemistry and biology (Jia et al., 2021; Alber et al., 2019) to problems in applications such as robotics (Collins et al., 2021; Narang et al., 2022) or computer vision

(Müller et al., 2018). Simulation offers the dual advantages of scalable and cheap data collection and an easy way to encode domain-specific prior knowledge into end-to-end machine learning problems. This is particularly important for data-scarce problems such as robotics, where collecting real data can lead to costly and unsafe failures or may require expensive human supervision. Critical to each of these endeavors is a rich and accurate simulation environment, complete with complex scene layouts and kinematic structure. For instance, advances in robotic mobile manipulation in the Habitat simulator (Szot et al., 2021), are critically dependent on the Matterport dataset for realistic scenes (Yadav et al., 2023). The creation and curation of these simulation scenes is an often overlooked part of the process.

The de-facto process for generating simulation content is either manual (Kolve et al., 2017) or procedural (Deitke et al., 2022). The manual process for creating simulation scenes involves the algorithm designer working to characterize, identify, and model a particular real-world scene, a painstaking and impractical process. This leads to content that is not very diverse due to the onerous human effort required. On the other hand, rule-based procedural generation methods (Deitke et al., 2022; Raistrick et al., 2023) have seen success in particular machine learning applications such as embodied navigation, but often struggle to capture the natural complexity of the real world. Moreover, the procedural generation process is not controllable, making it hard to generate simulation content corresponding to a *particular* real-world environment. The inability of the current status quo in the generation of simulation content - both procedural generation and manual creation, makes apparent the necessity of a targeted technique for scalable content creation in simulation that is able to retain realistic kinematic and semantic structure.

What are the desiderata for such a content creation method? To enable a variety of downstream use cases, scalable content creation in simulation must be (1) realistic enough such that inferences made in simulation transfer back to the real world (2) diverse in a way that captures natural statistics so as to enable learning generalizable models and policies (3) controllable in a way that allows for targeted generation of particular scenes of interest. While a variety of methods for scene generation and inverse graphics (Kulkarni et al., 2015; Lunz et al., 2020; Jaques et al., 2020) satisfy one or more of these criteria, to the best of our knowledge, it has proven challenging to develop content creation methods that satisfy them all. To generate content of this nature, we develop methods that map directly from isolated real-world images to corresponding simulation content (expressed as a Unified Robot Description File (URDF)) that could plausibly represent the semantics, kinematics, and structure of the scene. This is an inverse mapping problem going from real-world images to kinematically accurate, interactive simulation. While inverse modeling problems in the literature have been tackled with data-driven techniques such as supervised learning, in this case, a large-scale paired dataset of realistic images and their corresponding simulation environments does not readily exist in the literature.

Our key idea is that we can generate a suitable dataset for inverse modeling from images to plausible simulations by leveraging controllable text-to-image generative models (Rombach et al., 2022). From a set of procedurally or manually constructed scenes, we can generate realistic images that are representative of that particular simulation scene. This paired dataset of simulation scenes and corresponding realistic images can then be *inverted* via supervised learning to learn a model that maps from realistic images directly to plausible simulation environments. This learned model can generate realistic and diverse content directly from real-world images mined from the web without any additional annotation. The resulting models can be used in several use cases - (1) diverse generation: generating a large and diverse set of realistic simulation environments that correspond directly to real-world images, or (2) targeted generation: generating a simulation environment (or narrow distribution of environments) corresponding to a particular set of desired images.

## 2 RELATED WORK

This work is related to a large body of work in inverse-graphics, procedural generation, 3-D reconstruction and data augmentation. We provide some context on these related fields below.

**Inverse-Graphics:** Inverse graphics is a well-studied and rich field of study looking to infer the properties of a scene from images (or videos) of the scene of interest. A variety of work focuses on inferring scene properties such as geometry, lighting, and other geometric properties from single images (Battaglia et al., 2013). This work has both been optimization-based (Agarwal et al., 2011) and learning-based(Park et al., 2019). In a similar vein, a rich body of work (Samavati & Soryani,

2023) focuses on mesh reconstruction and novel view synthesis using a variety of techniques such as implicit neural fields (Mildenhall et al., 2021; Park et al., 2020; Zhang et al., 2020), Gaussian splatting (Kerbl et al., 2023; Luiten et al., 2023), differentiable rendering (Nguyen-Phuoc et al., 2018; Kato et al., 2020; Liu et al., 2018) amongst many other techniques. Importantly, the focus of many of these works on inverse graphics has been on geometric reconstruction rather than our focus on scene-level simulation construction complete with kinematic and semantic structure like object relationships and articulation. There have been a number of efforts in inferring physical properties such as articulation (Xu et al., 2019; 2022; DeMoss et al., 2023), friction and surface properties (Wu et al., 2017; Piloto et al., 2022; Kubricht et al., 2017; de Avila Belbute-Peres et al., 2018), although these typically require either interaction or video access. In contrast, our work focuses less on exact geometry reconstruction but rather on generating correct scene statistics at the articulation/kinematics/positioning level for entire scenes or complex objects from single RGB images. As opposed to these methods, the goal is not just a slow and expensive process for a single scene, but a fast generation process that can scale to generate hundreds of scenes with natural statistics. Importantly, this generation process does not require interaction or targeted data collection per domain.

Generating indoor scenes is a long-standing problem in computer vision and machine learning. This has been approached by building learned generative models of indoor scenes (Ritchie et al., 2019; Li et al., 2019; Keshavarzi et al., 2020; Hudson & Zitnick, 2021) and floorplans (Hu et al., 2020; Nauata et al., 2021; Wang et al., 2021), while others have produced text-to-scene models (Chang et al., 2014; 2015). While generating scenes this way can be promising, these methods either fail to achieve the targeted generation of complex scenes with articulation and complex kinematic structure intact or require extremely expensive inference processes to do so. On the other hand, procedural generation techniques have been popular in generating grid-world environments (Khalifa et al., 2020; Earle et al., 2021; Dennis et al., 2020; Gisslén et al., 2021) and in generating home environments at scale (Deitke et al., 2022). These scenes are diverse and often rich, but are not controllable to particular target scenes or are not able to generate scenes complete with physical properties and articulation. Other techniques such as (Li et al., 2021; Deitke et al., 2023) are able to generate large datasets of more interactive scenes but require interactive scanning with either a phone or other hardware for dataset generation specific to indoor scenes. URDFormer is able to generate realistic, diverse, and controllable scenes while retaining rich kinematic and semantic structure from internet images alone.

**Data Augmentation with Generative Models** Our work is certainly not the first (Eigenschink et al., 2023) to use synthetic data generated by generative models for training networks that can then be deployed on real data. These models have been used the context of data augmentation (Chen et al., 2023; Yu et al., 2023; Trabucco et al., 2023), representation learning via self supervised learning (Fu et al., 2023; Tian et al., 2023; Jahanian et al., 2022), model selection (Shoshan et al., 2023) and even applications like healthcare (Choi et al., 2017). In contrast to these works, our work shows that controllable generative modeling can be used to generate datasets that are suitable for inverse modeling for creating simulation assets at scale.

## 3 URDFORMER : GENERATING INTERACTIVE SIMULATION ENVIRONMENTS BY LEARNING INVERSE MODELS FROM GENERATED DATASETS

Generating simulated scenes with a high degree of visual realism that supports rich kinematic and dynamic structure is a challenging problem. Downstream applications in robotics and computer vision typically require data that is both **realistic**, **diverse**, and **controllable**. To accomplish these requirements, we take an inverse approach to the problem and generate scenes by mapping real RGB images to scene representations complete with kinematics and semantics. This allows for scene generation that is **realistic** since it inherits natural scene and object statistics from real images. The generated scenes are **diverse** since large image datasets with diverse content can be used to seed such a generation process. Lastly, the generation is **controllable** since curated images of particular target environments can be used to generate corresponding simulation assets. We first define the inverse problem of synthetic scene generation from real-world images, then describe how to learn inverse models to solve this problem with supervised learning on a paired dataset generated using pre-trained controllable generative models. Finally, we show how the learned inverse model can be used with real-world image datasets for scalable content creation.

## 3.1 PROBLEM FORMULATION

To formalize the problem of simulation scene generation from real-world images, let us consider a kinematic scene description $z$ drawn from a target scene distribution $P(z)$ in the real world. For our purposes, the scene can be described as a list of objects $z = \{o_1 \ldots o_n\}$, where each object $o_i$ contains a class label $c_i$, a 3D bounding box $b_i \in \mathbb{R}^6$, a 3D transform $T_i \in SE(3)$, a kinematic parent that references a previous object $p_i \in [1 \ldots i-1]$ and a joint type $j_i$ that specifies how that object can move relative to its parent $o_i = (c_i, b_i, T_i, p_i, j_i)$. The kinematic structure $z$ for a particular real-world scenario is unknown without extensive human labeling effort, and instead, we only have access to the result $x$ of an indirect "forward" function $f$, $x = f(z)$. For example, $x$ could be a photograph of the real environment, or a point cloud captured with a LIDAR scanner. The goal in this work is to recover the entire kinematic and semantic structure of the scene, thereby requiring complete inference of a rich scene representation $z$.

Unfortunately, since the content $z$ is unknown for most complex real-world scenes and difficult to generate manually, it is challenging to solve the "inverse" generation problem to infer the scene description $z$ from the forward rendered images (or alternative sensor readings) $x$, $z = f^{-1}(x)$. Had there been a sizeable dataset $\mathcal{D} = \{(z_i, x_i)\}_{i=1}^N$ of scene descriptors $z_i$ in simulation and their corresponding real-world counterparts $x_i$, the inverse problem could have been solved using supervised learning (minimizing a loss $\mathcal{L}$ like the cross entropy loss or a MSE loss) to learn an $f_\theta^{-1}$ that predicts the scene descriptors $\hat{z}$ given an input forward-rendered image $x$.

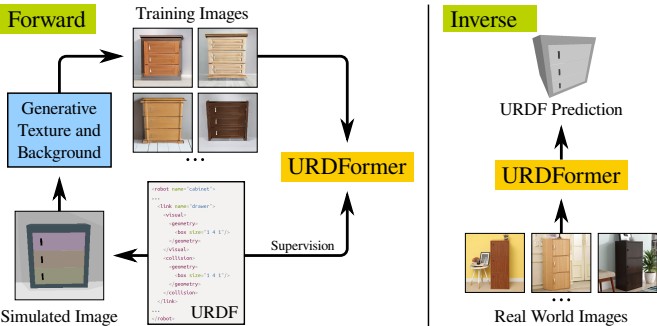

Figure 2: An overview of the training and application of URDFormer . During the forward process, existing simulation assets are first used to generate a large paired dataset of simulation assets and realistic rendered images. This paired dataset is used to train the URDFormer inverse model that can predict URDFs from RGB images. This model can then be used with real-world images to generate novel simulations.

However, this is challenging due to the scarcity of the dataset. To circumvent this issue, we leverage pre-trained generative models that convert procedurally generated scenes in simulation into a large paired dataset of scene content $z$ and their corresponding realistic images $x$. This process can generate a large and diverse dataset of image and scene-description $(x, z)$ pairs that we can use to train an effective inverse model $f_\theta^{-1}(x)$ for generating scene descriptions $\hat{z}$ from real RGB images $x$. Since most scenes that we consider are object-centric, we decompose the inverse problem into two parts: (1) object-level prediction that focuses on the kinematic structure of individual objects, and (2) Global-scene Prediction that focuses on the structure of an overall scene. We next discuss the process of generating a large paired dataset for these two components and then show the training process for the inverse model in detail.

## 3.2 CONTROLLED GENERATION OF PAIRED DATASETS WITH GENERATIVE MODELS

Given a simulated scene $z$ (drawn from a dataset such as (Mo et al., 2019), or procedurally generated), we use the fact that controllable generative models are both diverse and realistic enough to take an unrealistic rendering of a scene in simulation and generate a distribution of corresponding *realistic* images. This allows the scene in simulation with unrealistic appearance and texture to be translated into a diverse set of visually realistic images that plausibly match the same underlying environment. To ensure piecewise consistency and realism of the generated images, we use two different dataset generation techniques for the scene structure and object structure respectively. These share the same conceptual ideas but differ to account for consistency properties in each case.

**Scene-Level Dataset Generation:** To generate training data for the scene model, we feed the rendered image from simulation along with a templated text prompt to an image-and-text guided diffusion model (Rombach et al., 2022). This model generates a new image that attempts to simultaneously match the content described in the text prompt while retaining the global scene layout from the

provided image. We found that this model is able to reliably maintain the scene layout, but it may change some individual components of the scene, for example replacing objects with a different but plausible category, or changing the number of components under an object such as the drawers or handles. Despite these failures, the large-scale structural consistency still provides a useful source of training data. After running our simulated image through the generated model, we have realistic images that contain known high-level object positions and spatial relationships, but unknown category and low-level part structures. This means that the scene model dataset contains complete images, but incomplete labels. Rather than complete $(x, z)$ pairs, we have a dataset $\mathcal{D}_{\text{scene}} = \big\{(x, \tilde{z})\big\}$ of $(x, \tilde{z})$ pairs where $\tilde{z}$ only contains the bounding boxes, transforms and parents of the high-level (non-part) objects $\tilde{z} = \big\{(b_1, T_1, p_1) \dots (b_n, T_n, p_n)\big\}$.

**Object-Level Dataset Generation:** The process for generating object-level training data is similar, but requires more care due to the tendency of generative models to modify low-level details. For objects with complex kinematic structure, such as cabinets, we procedurally generate a large number of examples of these objects and render them in isolation from different angles. Rather than using the generative model to construct entirely new images, we use it to produce diverse texture images, which are overlaid in the appropriate locations on the image using perspective warping. We then change the background of the image using the generative model with appropriate masking derived from the original render. For less complex objects that do not have important part-wise structure, we simply replace the rendered image with a new sample from the image-and-text guided generative model. Unlike the scene dataset which contains complete images but partial labels, the object dataset contains partial images in the sense that they contain only a single object, but complete labels for the object and its kinematic parts. We can say that this dataset $\mathcal{D}_{\text{object}}$ contains $(\tilde{x}, z)$ pairs where $\tilde{x}$ is an image of a single object rather than a full scene (hence the partial $x$), and $z$ is complete for the single object and its parts. The result of these two data generation processes is a high-level scene structure dataset $\mathcal{D}_{\text{scene}}$, and a low-level object dataset $\mathcal{D}_{\text{object}}$.

## 3.3 URDFORMER : LEARNING INVERSE GENERATIVE MODELS FOR SCENE SYNTHESIS

Given the datasets $\mathcal{D}_{\text{object}} = (\tilde{x}, z)$ and $\mathcal{D}_{\text{scene}} = (x, \tilde{z})$ constructed as described above, we can use supervised learning methods to learn an *inverse model* that maps images of a complex object or scene to the corresponding simulation asset. In order to take advantage of these partially complete datasets, we must add some structure to our prediction model. We do this by splitting our learned inverse model in correspondence with the split in our forward model: we train one network $f_\theta^{-1}$ to predict the high-level scene structure using dataset $\mathcal{D}_{\text{scene}}$ and another network $g_\phi^{-1}$ to predict the low-level part structure of objects using $\mathcal{D}_{\text{object}}$.

To model both the scene-level prediction model ($f_\theta^{-1}$) and the low-level part prediction model ($g_\phi^{-1}$), we propose a novel network architecture - URDFormer, that takes an RGB image and predicts URDF primitives as shown in Figure 3. Note that both the scene-level prediction and the low-level part prediction use the same network architecture, the scene-level simply operates on full images with object components segmented, while the part-level operates on crops of particular objects with parts segmented. In the URDFormer architecture, the image is first fed into a ViT visual backbone(Dosovitskiy et al., 2020) to extract global features. We then obtain bounding boxes of the objects in the image using the masks rendered from the original procedurally generated scene in simulation (these are known at training time, and can be easily extracted using segmentation models at test time). We then use ROI alignment (He et al., 2017) to extract features for each of these bounding boxes. These feature maps are combined with an embedding of the bounding box coordinates and then fed through a transformer (Vaswani et al., 2017) to produce a feature for each object in the scene. An MLP then decodes these features into an optional class label (used only when training the object-level model), and a discretized 3D position and bounding box. In addition, it also produces a child embedding and a parent embedding that are used to predict the hierarchical relationships in the scene (object to its parent and so on). To construct these relationships, the network uses a technique from scene graph generation (Yang et al., 2023) that produces an $n \times n$ relationship score matrix by computing the dot product of every possible parent with every possible child. The scene model also has learned embeddings for six different root objects corresponding to the four walls, the floor, and the ceiling so that large objects like countertops and sinks can be attached to the room.

Due to the unpredictable nature of the generative transforms that are used to make the scene image realistic, which may change class identities, only the position, bounding box, and relationship information is used when computing the high-level scene structure. To compute the class labels for

the top-level objects, we use max-pooling of the dense ViT features along with an MLP in the part-prediction model $g_\phi^{-1}$. To generate a full estimate of the scene description from a natural image at test time, the image and a list of high-level bounding boxes are first fed to the scene prediction model $f_\theta^{-1}$, which predicts the location and parent for each object. The image regions corresponding to these boxes are then extracted and further segmented to produce part-level bounding boxes. Each of these image regions and the corresponding part boxes are then fed into the part prediction model to compute the kinematic structure of the low-level parts. This nested prediction structure can be used to generate entire scenes from web-scraped RGB images drawn from any image dataset to generate novel simulation content both at the scene level and at the object level.

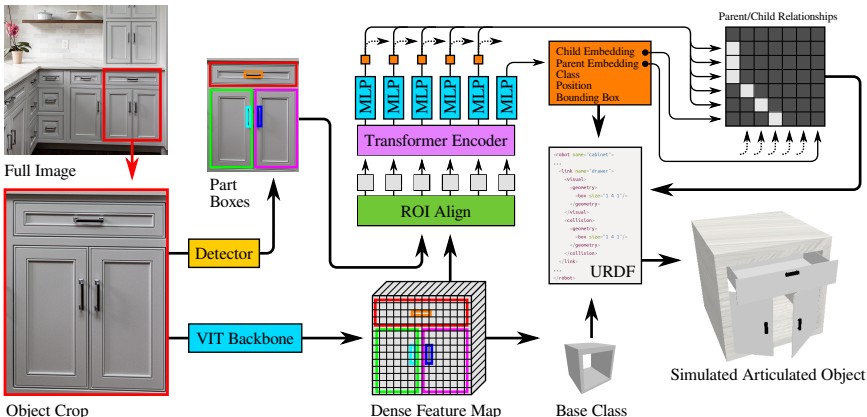

Figure 3: Architecture of URDFormer : an inverse model (URDFormer ) that predicts simulation parameters from RGB images. URDFormer can translate web-scraped real-world RGB images of scenes into complete simulation assets. The model shown here is used to estimate the part structure of an individual object. When estimating the scene structure, the Object Crop image would be replaced by an image of the entire scene.

## 4 EXPERIMENTS

In this section, we perform an empirical evaluation to answer the following questions: (1) Can the dataset generation method introduced in section 3.2 generate consistent realistic images for simulation content? (2) Is URDFormer able to generate plausible and accurate simulation content from novel real-world images? (3) Which elements of the structured prediction pipeline are most important in ensuring accurate simulation generation?

### 4.1 PHASE 1: (FORWARD) PAIRED DATASET GENERATION

To synthesize the initial paired dataset, we first procedurally generate a set of URDF representations of scenes in simulation both for global scenes like kitchens and for single objects like fridges, cabinets, and drawers. These initially generated simulation scenes are shown in Fig5 (Left). We can then follow the procedure outlined in Section 3.2 for the controlled generation of paired datasets to generate a large dataset of simulation scenes and paired realistic RGB images as shown in Fig5 (Right) (More visualizations and videos are available on the website). For objects with diverse parts, we observe that depth-guided stable diffusion (Rombach et al., 2022) often ignores the semantic details of local parts, leading to inconsistency issues shown as Fig 7 in Appendix A.1. To overcome this issue, we use images of texture to guide diffusion models to generate large and diverse texture templates and randomly choose one template and warp it back to the original part region using perspective transformation. We apply in-painting models for smoothing the boundary of the parts and generating content for the background. We visualize this process in Fig 4. In total, we generated  260K images for global scenes of kitchens and living rooms, and  235K images of 14 types of objects such as cabinets, ovens, and fridges. Details of the dataset can be found in Appendix B.1.

### 4.2 PHASE 2: (INVERSE) REAL-WORLD URDF PREDICTION

Given the generated paired dataset shown in Fig 5, we next evaluate how successful a trained inverse model is at generating simulation scenes representing unseen real-world test images.

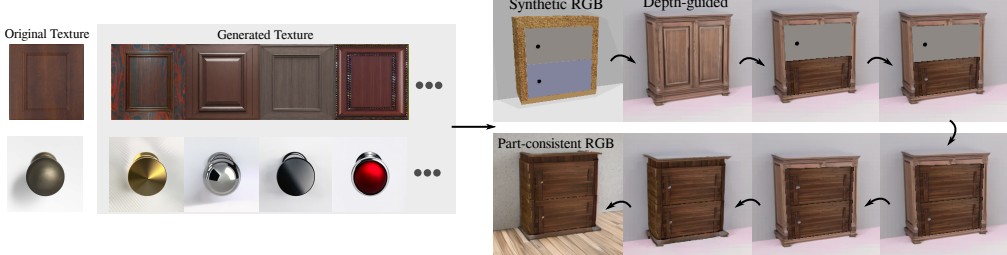

Figure 4: Paired dataset generation using texture and prompt templates to guide Stable Diffusion (Rombach et al., 2022) and create a diverse texture dataset, which can be then warped on the targeted individual part of the object, as described in Section 3.2

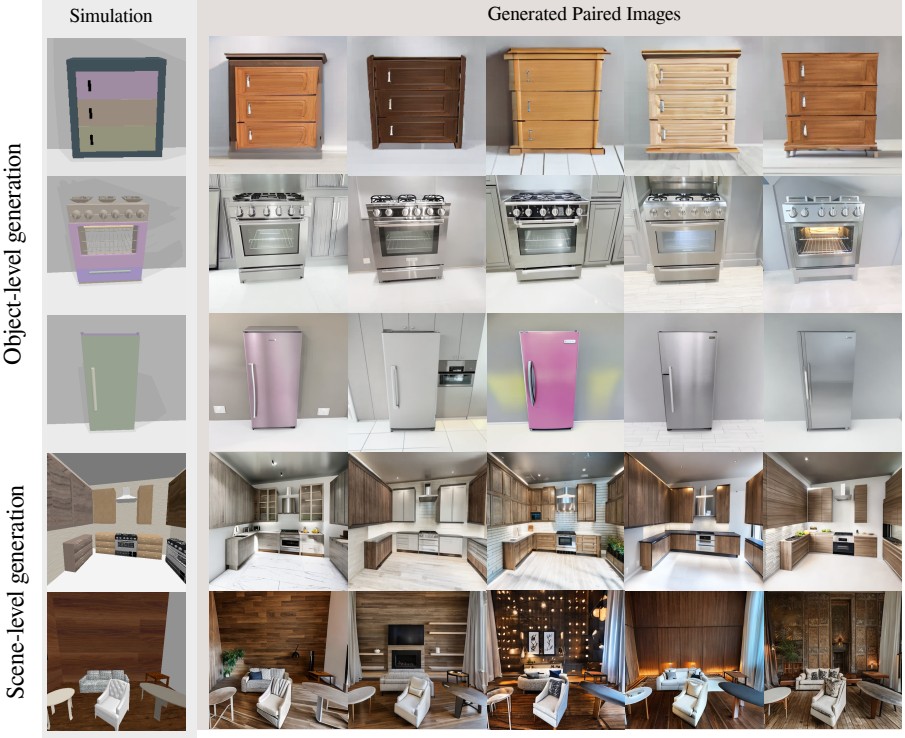

Figure 5: Qualitative results on (forward) paired dataset generation. Left: Original simulation images. Right: Generated realistic images that match the URDF descriptions of the scene on the left.

**Real World Dataset:** We create two types of datasets for evaluation: (a) Obj300 includes URDFs of 300 internet images of individual objects from 5 categories including 100 cabinets, 50 ovens, 50 dishwashers, 50 fridges and 50 washing machines. (b) Global scenes include URDFs of 80 internet images including 54 kitchens and 26 living rooms. For each scene, we manually label the bounding box for each object and its parts, as well as the URDF primitives including mesh types, parent bounding box ID, positions, and scales relative to its parent. We use the mesh types such as "left door", and "right door" to infer link axis and joint types. All the position values and scale values are discretized into 12 bins.

**Evaluation Metrics:** Evaluating entire scenes is challenging given the mixed structure and subjective nature of human labelling. We adopt an edit-distance based metric for structural comparison, and use a small dataset of manually labelled examples for evaluation.

(1) Edit Distance with Bounding Box Offset: We evaluate our predicted scene structure using a tree edit-distance metric. This method requires access to a predicted and ground-truth kinematic tree. We start at the root of the kinematic tree and use the Hungarian method to compute the lowest-cost assignment between the children of the predicted root and the children of the ground truth root where the cost is based on their spatial coordinates. If there are more predicted children than ground truth,

Table 1: Comparison with baseline methods: trained with random colors, selected textures, and random textures, as well as prompt guided BLIP2. URDFormer with generated realistic textures predicts more accurate simulation content from unseen real-world images.

| | Obj300 ($\downarrow$) | Global (Obj) ($\downarrow$) | Global (Parts) ($\downarrow$) |
|---|---|---|---|
| URDFormer (Random Colors) | 1.08 | 10.81 | 19.62 |
| URDFormer (Selected Textures) | 0.63 | 9.87 | 19.11 |
| URDFormer (Random Textures) | 1.22 | 11.85 | 18.67 |
| Guided BLIP2 | 4.27 | 14.64 | 24.58 |
| **URDFormer (Generated Textures (ours))** | **0.42** | **9.51** | **18.21** |

Table 2: Ablation study on training with different vision backbones and input features, showing training using both visual/spatial features, with a backbone pretrained on diverse real images achieves higher performance.

| | ED Box ($\downarrow$) | ED IoU$_{0.25}$ ($\downarrow$) | ED IoU$_{0.5}$ ($\downarrow$) | ED IoU$_{0.75}$ ($\downarrow$) |
|---|---|---|---|---|
| Scratch | 7.00 | 6.15 | 8.37 | 14.48 |
| Pretrained on ImageNet | 6.33 | 5.48 | 7.74 | 13.85 |
| **Pretrained MAE** | **5.70** | **5.11** | **7.07** | **13.41** |
| Pretrained MAE (No bbox) | 6.19 | 5.26 | 7.63 | 14.11 |
| only with bbox | 7.04 | 6.52 | 8.26 | 14.26 |

the unassigned predicted children and all of their descendants are marked as **False Positive** edits. Conversely, if there are more ground truth children than predicted children, the unmatched ground truth children and all of their descendants are marked as **False Negative** edits. We then compare the spatial parameters of the matched predicted and ground truth children. If they are not close enough to each other according to a fixed threshold, the predicted child and its descendants are marked as **False Positives**, and the ground truth child and its descendants are marked as **False Negatives**. If the two are close enough, the class label of the predicted child is compared against the class label of the ground truth child. If they do not match, we add a **Class Incorrect** edit. Regardless of whether the classes match, this process is recursively applied to the matching children. To compute a single score, we assign weights to these edits based on their position in the hierarchy and sum them. For the experiments in this paper, we assigned a weight of 1.0 to all edits at the top level corresponding to objects, a weight of 0.5 to the parts such as cabinet doors, and a weight of 0.1 to all objects further down the hierarchy such as handles and knobs attached to doors.

(2) Edit Distance with IoU: Similar to bounding box offset, we simply replace the spatial coordinate cost with IoU between two bounding boxes. We define levels of threshold based on overlapping areas: ED IoU$_{0.25}$, ED IoU$_{0.5}$, ED IoU$_{0.75}$. We show evaluation using both metrics in ablations, but in general, we found the two metrics yield the same performance, thus we only use edit distance with a bounding box for baseline evaluation.

**Baselines** We compare URDFormer against several other baselines in Table 1. In particular, to show the importance of pixel realism, we compare with training on (1) Random Colors (2) Selected Realistic Textures (3) Random Textures (Visualizations of their differences are in Appendix A.2). In addition, we also compare our method against recent Vision-Language Models with guided prompts: Guided BLIP2. In particular, (1) Random Colors randomly selects RGB values for each part inside the scene and (2) Selected Realistic Textures manually selects texture images for corresponding objects. (3) Random Textures selects random images. (4) Guided BLIP2 takes a sequence of question prompts and guides pretrained BLIP2 models Li et al. (2023) to output the URDF primitives in the valid format (Please check Appendix C.1 for prompt details). We observe that training with generated realistic visual features improves the generalization to real-world images. Although trained on large real-world datasets, BLIP2 fails to reason about the 3D structure of the scene as well as the kinematics structure of individual objects, showing using a more structured and targeted dataset is important during training. Here Global (Obj) represents the evaluation of high-level position/parent reasoning, while Global (Parts) represents the evaluation of the full scene including the high-level and detailed kinematic structure of each object.

**Ablations** To study how different components of URDFormer impact the performance, we perform an ablation study on (1) Do backbones pretrained on real-world images help with generalization?

(2) What are the important features of learning 3D kinematic structures, as shown in Table 2. In particular, we train URDFormer with three types of backbones: (1) vit-small-patch16-224 trained from scratch (2) finetune vit-small-patch16-224 pretrained on ImageNet (3) finetune vit-small-patch16-224 trained in (Radosavovic et al., 2023) on 197K kitchen scenes and evaluate on 54 real-world kitchen images. We observe that finetuning the vision backbone that is pretrained on real images performs better than training from scratch, and pretrained in (Radosavovic et al., 2023) achieves the best performance, which is likely due to the fact that it was trained on more diverse datasets than ImageNet. We observe that both training with only image features and training with only bounding box features decrease the performance, indicating the importance of both spatial and visual features.

**Qualitative Results:** We show the qualitative results of our URDF predictions in Fig 6. We use the same color to represent the same mesh type for better visualization. We observe that training with data generated using the method described in section 3.2 provides diverse visual information compared to baseline methods such as random colors or random textures. This is important for distinguishing mesh types such as stove and fridge, and reasoning about structure relations such as "cabinet on the right" and "cabinet in the front".

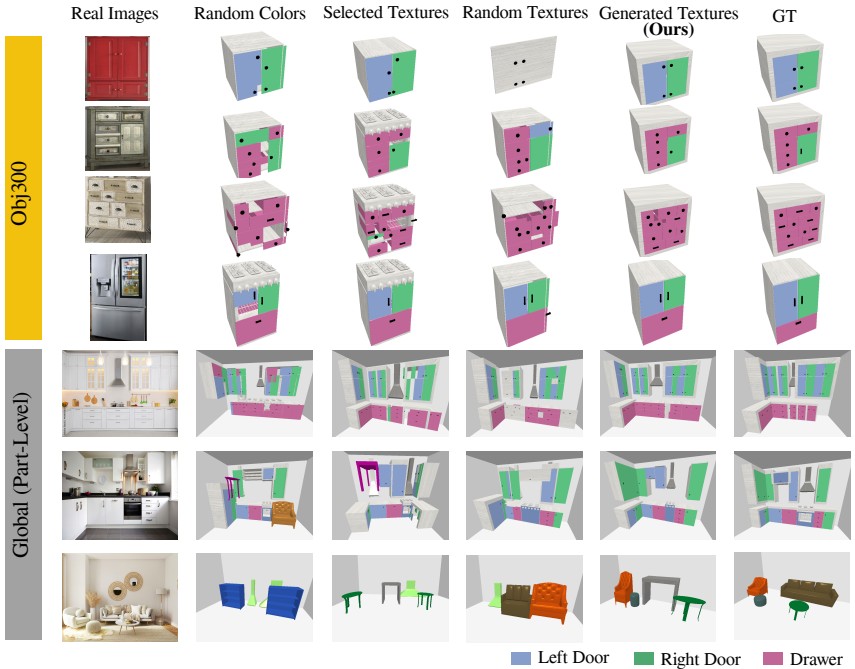

Figure 6: Evaluations of generated simulations on unseen real-world images. The left-most column indicates the real-world image input and each column indicates the performance of an inverse URDF prediction model trained with different training sets. We evaluate training datasets generated using random colors, selected textures, random textures, and textures generated with pre-trained generative models (ours), and compare these with ground truth URDF labels.

## 5  DISCUSSION

In this work, we presented URDFormer , a general-purpose, scalable technique for generating simulation content from real-world RGB images. We first generate a large-scale paired dataset of procedurally generated simulation content and a corresponding realistic RGB image using pre-trained controllable generative models. We then use our generated paired dataset to train an inverse model that maps directly from single RGB images to corresponding representations of scenes or complex objects in simulation. This inverse model can then be used with large image datasets of real-world RGB images to scalably generate simulation data complete with kinematic and semantic structure, without requiring any hand-crafting or hand-designing of these simulation assets. We show in our experimental results the efficacy of this scheme in generating assets at scale from real-world datasets of RGB images.

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

APPENDIX

## A  (FORWARD) DATA GENERATION

### A.1  PART-CONSISTENCY

We compare our part-wise generation method with other approaches qualitatively in Fig 7. In particular, we observe that depth-guided or in-painting stable diffusion models Rombach et al. (2022) often ignore local consistency, making it difficult to render high-quality images that are paired with the simulation content.

Original Sim RGB      Depth-guided      Inpainting      Structure-Aware (**ours**)

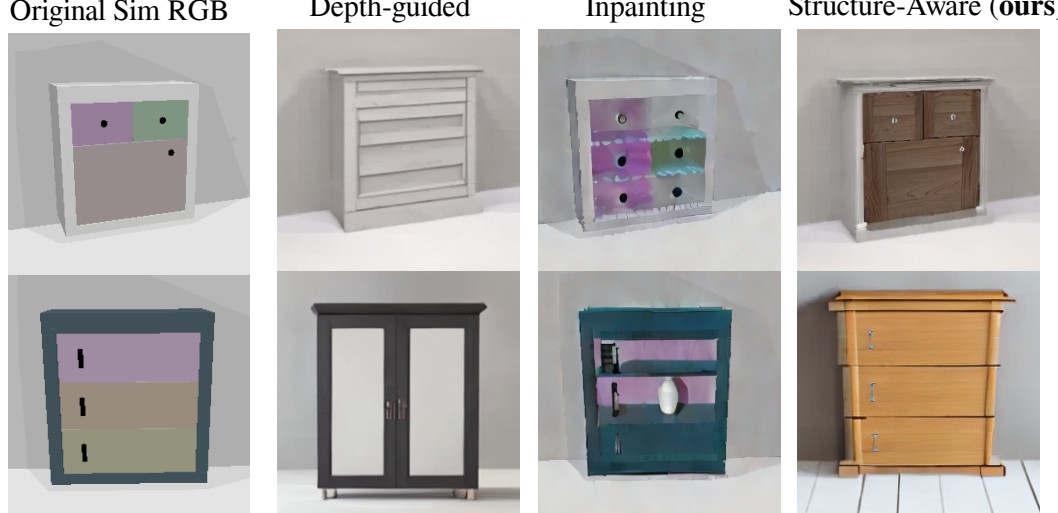

Figure 7: Qualitative comparison among different rendering methods: depth-guided diffusion models, inpainting stable diffusion and part-wise generation

### A.2  BASELINE DATA

We visualize the different training data for baseline methods shown in Table 1: URDFormer with random colors, selected textures, random textures, and generated textures. All baseline inputs are captured from the same camera angles. As shown in Fig 8, the generated texture shows high pixel realism that is closer to the distribution of the real world. As shown in Table 1, training on such data improves performance in predicting URDF structures from real-world images during the test time.

## B  TRAINING DETAILS OF URDFORMER

### B.1  DATASET

Our training dataset includes 267K global scene labels (197K kitchen scenes and 70K living room scenes) and 235K objects, which include 14 types of objects including cabinet, oven, dishwasher, washer, fridge, oven fan, shelf, tv, sofa, chair, square table, ottoman, coffee table and stuffed toy. Among these objects, 5 categories are articulated: cabinet, oven, dishwasher, washer, and fridge. These articulated objects include part meshes in from 8 types: drawer, left door, right door, oven door, down door, circle door, handle and knob.

### B.2  TRAINING DETAILS

All baseline methods (URDFOrmer with random colors, selected textures and random textures) are trained on one A40 GPU with batch size of 256. All baselines are trained with an equal number of epochs and evaluated using the last checkpoint.

| Random Color | Synthetic Texture | Random Texture | Generated |
|---|---|---|---|

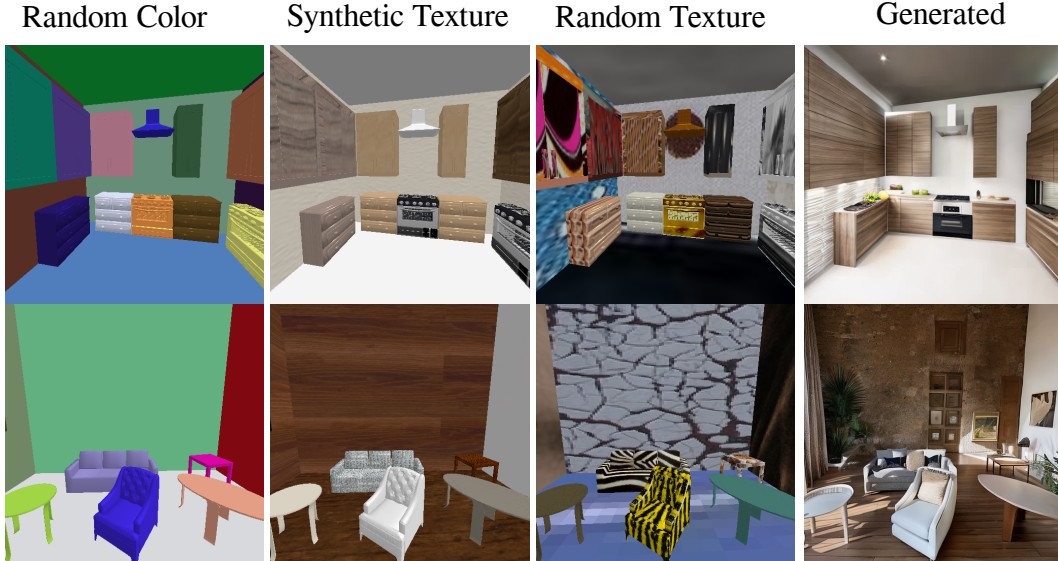

Figure 8: Comparison among baseline methods with different training input: Random Colors, selected textures, random textures and generated textures. Generated textures shows photo-realism that closer to the real-world distribution.

## C   EXPERIMENT DETAILS

### C.1   PROMPTS FOR BLIP2

In this section, we show examples of how we guide Vision-Language Models such as BLIP2 Li et al. (2023) to produce anwsers that can be converted into comparison results with ours.

**Global Parent Prompt:** "which of the wall is this object most likely on? choose one from 'floor', 'ceiling', 'front wall', 'left wall' and 'right wall'"

**Object Base Prompt:** "what's the name of the object. choose one word from cabinet, oven, dishwasher, washer, fridge, oven fan, shelf, tv, sofa, chair, square table, ottoman, coffee table and stuffed toy. Among these objects, 5 categories are articulated: cabinet, oven"

**Object Position Prompt:** "This image has a width of 512 and height of 512, the object box coordinate x is at 215, if the object scale is from 0 to 12, where do you imagine putting this bounding box relative to the object along the length in the 3D space. Choose from an integer from 0 to 12"

### C.2   COMPARE WITH OTHER SCENE GENERATION METHODS

We compare our pipeline with other methods of scene generation 9. In particular, we evaluate on (1) If the generated content follows the real-world structure (2) If the method works only on RGB images (3) if the method is fully automatic without human interaction with the scene (4) If it is scalable (5) if it can be applied to global scenes and (6) if the generated scenes are fully articulated.

| | Real World Distribution | RGB | Fully Automatic | Scalable | Scene Layout | Articulated Objects |
|---|:---:|:---:|:---:|:---:|:---:|:---:|
| Ditto | ✔ | ✘ | ✘ | ✘ | ✘ | ✔ |
| Ditto in the house | ✔ | ✘ | ✘ | ✘ | ✔ | ✔ |
| ProThor | ✘ | N/A | ✔ | ✔ | ✔ | ✘ |
| Phone2Proc | ✘ | ✘ | ✘ | ✘ | ✔ | ✘ |
| Ours | ✔ | ✔ | ✔ | ✔ | ✔ | ✔ |

Figure 9: Comparison against different approaches in scene generation: Ditto, Ditto in the house, ProcThor, Phone2Proc.

