# OpenReview forum: "URDFormer: Constructing interactive Realistic Scenes from Real Images via Simulation and Generative Modeling"
_ICLR.cc/2024/Conference — ICLR 2024 Conference Withdrawn Submission_

### Official Review · Reviewer_MNjq · 2023-10-16

**Soundness:** 4 excellent
**Presentation:** 4 excellent
**Contribution:** 2 fair
**Rating:** 3
**Confidence:** 3

**Summary:**

This paper provide a method which utilizes strong generative models to produce realistic training data based on the procedural object templates. It also design the URDFormer to predict the URDF from generated data, which allow us to predict URDF from more real-world images.

**Strengths:**

### Main Strength Points:

- Insightful observations from generative scenes and objects, which let us know the limitation of using generative models to produce data points. More specifically, it is about what information we can reuse from the model output and what cannot be reused.
- Many visualization results.
- Good topic, which will reduce human efforts to produce high-quality interactive data once well-studied.

**Weaknesses:**

#### Main Weakness Points:
- How do we get the bounding boxes and segmentation results for real-world images? It is straight-forward to use masks in the training data because the training data is procedural. But how do we train the detection and segmentation models for that, especially for the parts? What dataset is used for training these two models and how about the quantitative performance.visualization of these models?
- It is more like using data-augmentation (driven by generative models) to train a model for URDF prediction. However, as mentioned in the paper, we do not want the object hierarchy changes too much. That means the generative model will not do much more than **applying new generated textures** (from $D_{object}$) and **object replacement** (from $D_{scene}$). This will make the training data lacking of **part-level hierarchy variation**, which is the essential part of procedural generation. Therefore, I doubt its generalization capability on real-world images, especially for the objects which has different part-level hierarchy but still belong to the same category.

**Questions:**

Mentioned in weakness.

---

### Official Review · Reviewer_4K1q · 2023-10-29

**Soundness:** 3 good
**Presentation:** 3 good
**Contribution:** 2 fair
**Rating:** 6
**Confidence:** 3

**Summary:**

This paper proposes a pipeline of data generation, training, and prediction for inferring a simulatable scene structure from images. During the training stage, it first generates 3D models from scene description files (like URDF). The parameters of an object include its class label, 3D bounding box, 3D transformation, parent node, and joint type.

For both scene-level and object-level data generation, the method adds more diverse textures to the raw image through an image-and-text-guided diffusion model. One difference is that, on the object level, the generative model sometimes generates the texture image instead of the entire image. For prediction, the ViT-based network structures are similar for scene-level and object-level. The input is the image and segmentation, and the output is class labels, bounding boxes, and hierarchies.

Experiments show that this method can predict the simulatable config file from real-world images.

**Strengths:**

The experiment setting of this paper is good, as it validates the forward generation, inverse prediction, and design choices. The supplementary website also provides a large number of examples that can demonstrate the distribution, diversity, and quality of the predictions.

3D reconstruction for both geometry and structure is hard in general, but this paper converts it into a classification + bounding box prediction, which is a lot easier.

This method utilizes the power of pre-trained models to generate a large dataset and then solve the inverse problem. It is a promising paradigm for solving such difficult problems.

**Weaknesses:**

From the shown experiments we can see that the predicted object parts are mostly boxes. Since this method is actually predicting the object category instead of reconstructing the geometry, the fidelity of the generated results is relatively low, especially for objects with more variety in shapes, like chairs. The proposed paper may explain and compare the differences between works like [1].

This method does handle the joints in an ad-hoc way. They use the mesh types such as ”left door”, and ”right door” to infer link axis and joint types. Joints should be decoupled from the geometry. Sometimes, right/left is not well-defined when there are multiple parts and the object is not axis-aligned. Moreover, how to distinguish sliding/hinge/ball joints could be a problem. Moreover, the joint axis is also an important property for simulation.

The generated results are seemingly okay but might not be simulatable in some cases. In the examples shown on the website, there are overlapping doors which can result in collisions. It might be better if there were some post-processing steps to improve the prediction quality.

[1] Deep Part Induction from Articulated Object Pairs. Siggraph Asia 2018

**Questions:**

Is the URDF dataset big enough to generate enough training data? Especially for real-world scenarios where a room can have a very complicated layout and diverse object types.

How does this method work for a more complex hierarchy? Most of the examples shown here have very shallow kinematic trees (i.e. one parent node + multiple leaves.)

Can authors show more interesting geometries than boxes? Like a chair, or a robot hand, which is the most common URDF object?

---

### Official Review · Reviewer_prqQ · 2023-10-30

**Soundness:** 2 fair
**Presentation:** 3 good
**Contribution:** 2 fair
**Rating:** 5
**Confidence:** 4

**Summary:**

This paper introduces an innovative approach to create interactive scenes from real-world images. Initially, it either procedurally generates 3D scenes or taps into existing object/scene datasets to collect data with URDF. Subsequently, the method employs pretrained 2D diffusion models to produce realistic 2D images for the scene/object. This process allows for the creation of a dataset consisting of paired scenes/objects and their corresponding realistic images. The paper then introduces a network architecture that predicts the related URDF file using just a single image as input. This network is trained on the aforementioned generated dataset.

**Strengths:**

1. The subject of generating a 3D scene from a singular image holds significant importance.

2. The high-level idea of initially creating pairs of scenes and realistic images and subsequently training an inverse model is interesting.

3. The employment of 2D diffusion model for generating corresponding is interesting.

**Weaknesses:**

1. While the proposed method generates the URDF of the scene, it lacks accompanying 3D mesh files. It would enhance the paper's value if the authors provided concrete examples illustrating the practical use of these generated URDF files in the absence of mesh files. Further, elucidating the benefits of augmenting the number of URDFs with concrete applications could strengthen the presentation.

2. The method's current design presupposes easy access to procedurally generated 3D scenes or 3D object datasets with kinematic structures. This assumption may not hold true, as these could represent the genuine bottlenecks in the approach. For instance, curating 3D articulated object datasets demands substantial human intervention. Procedurally generating diverse and plausible 3D scenes isn't a straightforward task either. How the diversity of these scenes and objects impacts the inverse module's performance? Additionally, a clearer exposition regarding how training scenes and objects are obtained in the experiments would be beneficial.

3. Given the challenges in acquiring 3D scenes and articulated objects with URDFs, the method's applicability seems confined to a narrow range of scene or object categories.

4. As per Table 1, employing "selected textures" already produces almost comparable results. This observation somewhat diminishes the perceived necessity and advantage of using 2D diffusion models.


5. The evaluation appears somewhat limited. Incorporating a broader range of baseline methods beyond the currently employed self-designed baselines would offer a more comprehensive assessment. For example, there is a line of work on predicting the kinematic structure of the articulated objects. They should also be discussed and fairly discussed.

**Questions:**

1. How do you evaluate the accuracy of the predicted axis position?

2. "Due to the unpredictable nature of the generative transforms that are used to make the scene image realistic, which may change class identities, .... To compute the class labels for the top-level objects, " Can you elucidate the approach used to supervise the training of class label prediction?

3. "We visualize this process in Fig 4. In total, we generated 260K images for global scenes of kitchens and living rooms, and 235K images of 14 types of objects such as cabinets, ovens, and fridges" Could you clarify the number of distinct scenes or objects that were either generated or utilized?"

---

### Official Review · Reviewer_fQM1 · 2023-10-30

**Soundness:** 3 good
**Presentation:** 3 good
**Contribution:** 2 fair
**Rating:** 3
**Confidence:** 4

**Summary:**

The paper proposes URDFormer - an algorithm that is able to generate Unified Robot Description Format (URDF) primitives from real RGB images. The authors state that interacting with the real environment is a difficult task that generally involves manual modelling for each object. To address this issue, they propose a model that generates URDF files directly from RGB images.

The pipeline starts with an URDF file that is rendered in 3D. A background and texture for each part is added using generative models. This is the forward problem. The backward problem translates an RGB image to URDF primitives by first cropping the object of interest and then estimate its individual parts and relationships. Finally, the URDF file is generated. The authors introduce two datasets, Obj300, consisting of 300 object-centered images and Global scenes, consisting of 80 images with kitchen and living room scenes. Each image is manually adnotated with object parts. The authors also introduce a tree edit distance metric, with bounding box offset or IoU prediction for evaluating the prediction. Favourable results compared to a prompted BLIP2 model are presented.

**Strengths:**

- novel problem modelling to directly generate URDF files from RGB images -- this could be used to better understand the scene and interact with it
- staightforward concept of adding random textures and backgrounds from a diffusion model to improve the training results
- diffusion-based method to augment the training dataset / result quality
- favourable results compared to BLIP2

**Weaknesses:**

### Summary ###
Apart from the engineering part (diffusion-based renderings), the core contribution appears to be the conversion of a hierarchical part-based segmentation to an URDF file (parent, location, articulation). I belive this is simply not good enough for a main track paper, especially given the quality and quantity of the comparisons and metrics -- [1*] being the main contender.


### Detailed issues ###

- limited testing data
    - a total of 380 web-captured images from two datasets (300 + 80) is compiled by the authors
        - furthermore, the accuracy of some dataset images is questionable, see Fig 6 lower right image - the pose of the objects is not accurate
    - lots of [large] parts datasets that could be used/compared against (e.g., [1*] -- closest to the scope of the paper, [2*], some from [6*])
- limited method comparisons
    - a pretrained BLIP2 is the only method tested and it is by far not the best one  -- see [4*] for a more recent overview, lots of multimodal LLMs that could be prompted in the same way, or even better, use a hierarchical part segmentator such as [7*]
- metric
    - novel metric introduced while a standard [hierarchical] part metric such as PQ / PartPQ [3*] or even just a mAP + mobility metric (see [1*]) can be used
- weak baselines
    - I would argue that it is trivial to assess that just rendering 3D object with single colors or fixed texture would yield a lower quality result compared to the generative-based variety; generative models have been used before to enhance semgentation capabilities, for example in part-based[5*] or more recently few shot image segmentation [6*] -- and arguably this paper is solving the same task
- limited novelty / experiments
    - arguably, it is easy to convert to an URDF format once the part boxes and hierarchy are estimated with an existing hierarchical part-based method, especially for the simiplified scenario considered by the authors (object-centric crop); there are even open vocabulary hierarchical segmetation methods that could be used as a baseline before the URDF conversion (i.e., [7*]); no attempt has been made in this regard.
    - disregarding the URDF ouptut, the problem can be modeled as a part-based domain adaptation from synthetic to real, which is what happens in [8*]
   - C.2 appendix -- a set of methods listed but summarly dismissed

___
[1*] Mao, Y., Zhang, Y., Jiang, H., Chang, A., & Savva, M. (2022). MultiScan: Scalable RGBD scanning for 3D environments with articulated objects. Advances in Neural Information Processing Systems, 35, 9058-9071. https://github.com/smartscenes/multiscan

[2*] Mo, K., Zhu, S., Chang, A. X., Yi, L., Tripathi, S., Guibas, L. J., & Su, H. (2019). Partnet: A large-scale benchmark for fine-grained and hierarchical part-level 3d object understanding. In Proceedings of the IEEE/CVF conference on computer vision and pattern recognition (pp. 909-918).

[3*]de Geus, D., Meletis, P., Lu, C., Wen, X., & Dubbelman, G. (2021). Part-aware panoptic segmentation. In Proceedings of the IEEE/CVF Conference on Computer Vision and Pattern Recognition (pp. 5485-5494).

[4*] Yin, S., Fu, C., Zhao, S., Li, K., Sun, X., Xu, T., & Chen, E. (2023). A Survey on Multimodal Large Language Models. arXiv preprint arXiv:2306.13549. https://github.com/BradyFU/Awesome-Multimodal-Large-Language-Models/tree/Evaluation

[5*]Eslami, S., & Williams, C. (2012). A generative model for parts-based object segmentation. Advances in Neural Information Processing Systems, 25.

[6*] Tan, W., Chen, S., & Yan, B. (2023). Diffss: Diffusion model for few-shot semantic segmentation. arXiv preprint arXiv:2307.00773.

[7*]Wang, X., Li, S., Kallidromitis, K., Kato, Y., Kozuka, K., & Darrell, T. (2023). Hierarchical open-vocabulary universal image segmentation. arXiv preprint arXiv:2307.00764. https://people.eecs.berkeley.edu/~xdwang/projects/HIPIE/

[8*]Liu, Q., Kortylewski, A., Zhang, Z., Li, Z., Guo, M., Liu, Q., ... & Yuille, A. (2022). Learning part segmentation through unsupervised domain adaptation from synthetic vehicles. In Proceedings of the IEEE/CVF conference on computer vision and pattern recognition (pp. 19140-19151).

**Questions:**

1. Why prompted LLMs instead of hierarchical part segmentation methods?
2. Have you tested on other hierarchical part-based baselines or datasets? See weaknesses for suggestions.
3. Can you add a standard part-based metric / mAP + mobility score?
4. Generation questions: have you tried depth __and__ structure-aware generation? No discussion on image resolution/texture quality impact. Any comment on this?